# 3D Cephalometric Normality Range: Auto Contractive Maps (ACM) Analysis in Selected Caucasian Skeletal Class I Age Groups

**DOI:** 10.3390/bioengineering9050216

**Published:** 2022-05-17

**Authors:** Marco Farronato, Giuseppe Baselli, Benedetta Baldini, Gianfranco Favia, Gianluca Martino Tartaglia

**Affiliations:** 1Department of Orthodontics, Faculty of Medicine, University of Milan, 20100 Milan, Italy; benedetta.baldini@unimi.it (B.B.); gianluca.tartaglia@unimi.it (G.M.T.); 2Department of Electronics, Information and Bioengineering, Politecnico Di Milano, 20133 Milan, Italy; giuseppe.baselli@polimi.it; 3Department of Interdisciplinary Medicine, Odontostomatology Unit, University of Bari “Aldo Moro”, 70121 Bari, Italy; gianfranco.favia@uniba.it; 4Facial Surgery and Dentistry Fondazione IRCCS Cà Granda, UOC Maxillo, Ospedale Maggiore Policlinico, 20142 Milan, Italy

**Keywords:** cephalometry, neural networks, digital dentistry, orthodontics, anatomy, head and skull

## Abstract

The objective of this paper is to define normal values of a novel 3D cephalometric analysis and to define the links through an artificial neural network (ANN). Methods: One hundred and fifteen CBCTs of Class I young patients, distributed among gender-adjusted developmental groups, were selected. Three operators identified 18 cephalometric landmarks from which 36 measurements were obtained. The repeatability was assessed through the ICC. Two-dimensional values were extracted by an automatic function, and the mean value and standard deviation were compared by paired Student’s *t*-tests. Correlation coefficient gave the relationships between 2D and 3D measurements for each group. The values were computed with the ANN to evaluate the parameters normality link and displayed by Pajek software. Results: The ICC assessed an excellent (≥0.9) repeatability. Normal values were extracted, and compared with 2D measurements, they showed a high correlation on the mid-sagittal plane, reaching 1.00, with the lowest 0.71 on the lateral plane. The ANN showed strong links between the values with the centrality of the go-sagittal plane compared to the rest. Conclusions: The study provides a set of 3D cephalometric values obtained by the upper and lower 95% CI for the mean divided into the developmental stage subgroups. The two-dimensional measurements showed variable concordance, while the ANN showed a centrality between the parameters.

## 1. Introduction

Cephalometrics is a crucial point of morphological diagnostic procedures to assess cranio-dento-facial features, growth, development, and treatment modifications [1]. Moreover, to diagnose and classify malocclusions, the measured values of cephalometric parameters are usually compared with the standard values.

The conventional analysis is performed manually or by the use of digital cephalometric tracings software on lateral, frontal, and axial X-ray projections. However, conventional cephalometric measurements have several inconveniences, including errors of projection and errors regarding the identification of the landmarks. Conventional radiographic techniques collapse a three-dimensional (3D) structure onto a two-dimensional (2D) plane. The resulting superimposition of the anatomical structures complicates image interpretation and landmark identification. These distortions could end up reducing the measurement accuracy or inter-operator reliability [2,3].

The introduction of cone beam computed tomography (CBCT) helps to avoid these problems, allowing the acquisition of 3D images [4,5,6]. CBCT acquisition is done using a cone-shaped X-ray beam to capture multiple images of the patient [2,7,8]. Three-dimensional acquisitions also permit to import and export individualized overlap-free reconstructions and digital imaging and communication in medicine (DICOM) data to and from other software applications [2,9,10]. A number of software programs are dedicated to managing and analyzing DICOM images derived from CBCT images for orthodontic purposes.

In order to diagnose and to provide an orthodontic treatment plan, the measured values of 2D cephalometric measurements are evaluated within the standard values given by previous studies [1]. However, no normal values of CBCT cephalometric tracings have been described yet for Caucasian subjects.

To approach the complexity of data analysis, the available technologies for data mining benefit from the advance in artificial neural networks (ANN). Among those, Auto Contractive Maps (ACM) allow basic improvements in both robustness of use in badly specified and computationally demanding problems and output usability and intelligibility. The great number of variables considered in cephalometry can complicate the comprehension of the correlations between the parameters; to solve this, we propose an innovative approach to the statistical analysis used in artificial intelligence systems. In particular, ACM allow the clinician to visualize the weighted maps of the “closeness” between the variable organizing them into a visible scenario [11]. The ACM system finds, by a specific learning algorithm, a square matrix of weighted connections among the variables of any dataset [12]. The weights matrix are then filtered by a minimum spanning tree (MST) algorithm, which generates a graph [11]. This model allows to unveil hidden associations and trends among a list of variables through a semantic connectivity map. The method displays the interactions of the relevant networks between and among the variables. Hubs are defined as the parameters presenting the maximum amount of connections in the resulting map. The specificity of the ACM algorithm allows to reduce a complex cost function [12].

Other systems have been deployed for the automatic individualization of points localization, with excellent results compared to the expert trained specialists; however, their use has never been suggested for the raw data analysis [13]. Specifically, we applied network-based analysis methods to the Class I measurements to “link” together the entities comprising that system and to find the relevant “strong connections” between normal values. The networks can be exported as *.net files, and a multitude of softwares can be used to visually display the connections in 2D, such as Pajek, UCINET, and NetDraw. The aim of this study is to propose a 3D cephalometric normal range as the baseline data for the cephalometric diagnosis of Caucasian patients. A set of CBCTs randomly taken from skeletal Class I patients were used in this scope. While artificial intelligence helps to understand the link between values in normal patients.

## 2. Materials and Methods

This comparative study was performed with Institutional Review Board approval of the Ethics Committee of the University of Milan (3 March 2016; n. 421).

### 2.1. Patient Selection

In total, 115 patients’ CBCTs were selected from a dataset of 700 full-head CBCT (field of view 20 × 25 minimum) of different patients who visited the Dental Department of the University of Milan from January 2010 to June 2020 for one of the following:-impacted and supernumerary teeth;-bicuspid tooth implant needs;-obstructive sleep disorders breathing and apnea syndrome;-orthognathic surgery;-trauma not involving mandibular or maxillary position;-foreign objects.

The patients’ CBCTs were selected following these inclusion criteria according to their anamnestic track:

patients with skeletal Class I (ANB angle between 0° and 4°, measured on the latero-lateral projection);

normal vertical dimension (Total Posterior Facial Height (TPFH) (S-Go)–Total Anterior Facial Height (TAFH) (N-Me) ratio between 60 and 64%) [14];

symmetry: a maximum difference of 3 mm between the midpoints of the right Gonion (rGo)–left Gonion (lGo) distance and the right Maxillary (rMx)–left Maxillary (lMx) distance in the posteroanterior projection [15]:no cross-bite;full dentition;absence of orthodontic appliances;absence of known craniofacial syndromes in the clinical history of the patient.

Before the CBCT scan analysis, all the selected patients and their parents were informed about the procedure and its risks and provided consent to the radiographic examination. They also permitted anonymized data use for research purposes. The study protocol was carried out according to the principles of the Helsinki Declaration, including all amendments and revisions.

### 2.2. Age and Sex Distribution

The criteria for age and sex distribution were addressed as follows:

In order to avoid gender-related development bias, patients were divided into males (*n* = 56) and females (*n* = 59) prior the division into the three developmental subgroups, according to Baccetti et al. (2002) for cervical evaluation, and Giannì (1980) for wrist evaluation:pre-growth peak (CS1-CS2) and (I-III period);growth peak (CS3) and (IV period);post-growth peak (CS4-CS5-CS6) and (V-VI period).

### 2.3. Scanning Protocol

All CBCT were recorded using the same machine with the same exposure parameters. Patients were placed in the same position, checked to ensure that their mouths were closed in habitual occlusion and instructed to remain still during the scan. Each scan was taken for 20 s at the 3.8 mA enhanced setting. The scans were then reconstructed at 0.3 mA. A 3D volumetric image of the patient was obtained using the iCAT ^®^ cone beam dental imaging system (1910 N. Penn Road, Hatfield, PA, USA). The scanning protocol involved a 4-mm slice thickness, a 16 × 22-cm field of view, a 20-s scan time, and a 0.49/0.49/0.5-mm voxel size. The scans were saved in Digital Imaging and Communications in Medicine (DICOM) format and transferred to a personal computer. The CBCTs were taken by the same expert technician.

### 2.4. Data Elaboration

The CBCT data were processed using Mimics software (version 22.0, Materialise, Leuven, Belgium), creating a set of cephalometric landmarks for the 3D–2D cephalometric analysis. First of all, a reference system was defined: a midsagittal plane (MSP) passing through landmarks S (Sella), N (Nasion), and Ba (Basion); a horizontal plane, perpendicular to MSP, through landmarks S and N; a frontal (coronal) plane, perpendicular to MSP; and the horizontal plane, passing through landmarks S and Me. Sella is intersected by the three different planes, and it is the center of the reference system (point 0, 0, 0).

### 2.5. 3D Cephalometrics

Eighteen cephalometric landmarks were defined using Mimics 22.0 (Materialise, Leuven, Belgium), according to the classical Steiner methods [15] (Figure 1).

Ten unpaired landmarks lying on the midsagittal plane:

N (Nasion), S (Sella), Ba (Basion), A (Point A), B (Point B), ANS (Anterior Nasal Spine), PNS (Posterior Nasal Spine), Me (Menton), UI (Upper Incisor), and LI (Lower Incisor).

Four paired landmarks divided into right and left:

Sor (Supra Orbital), Mx (Maxillar), Cd (Condylion), and Go (Gonion). A total of 36 measurements between them were automatically calculated by the function of the software measurements and analysis (21 linear, of which 7 were paired, unit: mm, and 15 angular, of which 5 were paired, unit: degrees):

Four Anteroposterior Measurements:

Anterior cranial fossa length (S-N): the distance between S and N;

Maxillary length (PNS-A): the distance between PNS and the A point;

Mandibular body length (right and left values: LGo-Me/RGo-Me): the distances between Go and Me.

Three Sagittal Angular Measurements:

SNA: the angle between landmarks S, N, and A, indicating the anteroposterior projection of the maxilla;

SNB: the angle between landmarks S, N, and B, indicating the anteroposterior projection of the mandible;

ANB: the angle between landmarks A, N, and B, indicating the anteroposterior inter-maxillary relationship.

Seven Vertical Linear Measurements:

Total anterior facial height (N-Me): the distance between N and Me;

Upper anterior facial height (N-ANS): the distance between N and ANS;

Lower anterior facial height (ANS-Me): the distance between ANS and Me;

Posterior facial height (right and left values: S-LGo/S-RGo): the distance between S and Go separately for the right and left sides;

Mandibular ramus height (right and left values: LCd-LGo/RCd-RGo): the distances between Cd and Go.

Twelve Vertical Angular Measurements:

Cranial base angle (Ba-S-N): the angle between Ba, S, and N;

Cranio-maxillary angle (S-N^^ANS-PNS^): the angle between the floor of the anterior cranial fossa and the palatal plane;

Cranio-mandibular angle (right and left values: S-N^^LGo/RGo–Me^): the angle between the floor of the anterior cranial fossa and the mandibular plane, measuring mandibular divergence;

Total gonial angle (right and left values: LCd/RCd—LGo/RGo—Me): the angle between the mandibular ramus and body;

Upper gonial angle (right and left values: LCd/RCd—LGo/RGo—N): can be used to predict mandibular growth;

Lower gonial angle (right and left values: N—LGo/RGo—Me): can be used to predict mandibular growth;

Divergence angle (right and left values: PNS/ANS—Go L/R—Me): can be used to measure the divergence between maxilla and the mandible.

Ten Transverse measurements:

Orbit distance (LSor/RSor—Midsagittal plane);

Condylar distance (LCd/RCd—Midsagittal plane);

Maxillary distance (LMx/RMx—Midsagittal plane);

Goniac distance (LGo/RGo—Midsagittal plane);

Upper dental symmetry (Ui—Midsagittal plane);

Lower dental symmetry (Li—Midsagittal plane).

Conventional 2D cephalometric measurements were obtained in the anteroposterior and laterolateral projections.

### 2.6. Data Reliability

The 3D and 2D cephalometric analyses were independently performed by three different experienced operators (>5 years of experience). All observers attended a calibration meeting aimed at making the measurements overlap and be reliable. All measurements were taken two times with a 2-week interval between each data collection.

### 2.7. Statistical Analysis

Sample size was calculated a priori with a two-sided Pearson’s chi-squared proportion test, the statistics power was set at 80% with an alpha of 0.05 and a delta of 0.351, and the effect size was set at a proportion of two groups: 0.5 pre-peak and 0.9 post-peak; the overall resulting sample size was a minimum of *n* = 50 cases; with an allocation of at least *n* = 25 per group (in a total of three groups), we increased the sample until reaching the uniformity between genders.

The collected data were statistically analyzed using IBM SPSS software. The mean value and standard deviation of each measurement were calculated separately for the 3D and 2D values and for each age subgroup. Standard Error Measurements (SEM) and 95% confidence intervals were calculated.

To evaluate the intra-rater reliability, the variations of the data measured by the same rater in the three observations under the same conditions, an Intra Class Coefficient (ICC) was calculated from a one-way random effects analysis of the variance model. To quantify the inter-rater reliability, the ICC was estimated after a multilevel mixed-effects linear regression among three raters. Mean estimations, along with 95% confidence intervals (95% CI), were reported for each ICC. The values of the intra- and inter-rater ICCs were interpreted according to Cicchetti and Sparrow [16]: [0; 0.40) poor repeatability, [0.40; 0.60) fair repeatability, [0.60; 0.75) good, and [0.75; 1.00] excellent repeatability.

As the data had a normal distribution, the mean values of the 3D and 2D cephalometric measurements were compared by Student’s paired *t*-tests. The significance level was set at 0.05. Finally, the resulting data were analyzed by ACM (AutoCM, Semeion, Italy), as previously described by Buscema et al. [12], to identify the betweenness centrality and authority node metrics. All the nonquantitative values were displayed as binary (for example, male/female 0/1). We set a value of 1000 epochs and a learning rate of 0.1. The resulting neural maps were exported as a file type *.net and displayed in a visual map calculator, Pajek, with the command: File > Network > Read (V 5.14, Operating system: Windows 11; http://pajek.imfm.si/doku.php. accessed on 10 November 2021.).

## 3. Results

In total, 53 pre-peak, 37 peak, and 25 post-peak individuals were included in the study.

Table 1 reports the results of the intra-rater reliability, and the data were aggregated from all the points measured. On all occasions, the ICCs were significant at *p* < 0.001. The same values were found for the inter-rater reliability. On all occasions, the values of the intra- and inter-rater ICC and relevant 95% CI showed an excellent repeatability and reliability. The lowest ICC was found on Left Cd–Go–N of 0.71 [16].

Table 2, Table 3 and Table 4 present the mean, SD, SEM, lower and upper limits of the 95% CI of the cephalometric values obtained using the 3D and 2D analyses, as well as the results of the correlation analysis between them for each age subgroup. The overall correlation between the 2D and 3D measurements was high for most of the points; the lowest observed correlation was for ANS-PNS–GoR-Me (0.71) and for CdL–GoL–Me (0.72) in the peak age subgroup. The overall normality range divided into the subgroups is presented in Table 5. The linear distances from the sagittal plane were excluded from comparative statistical tests, as they resulted, on average, as equal to the 2D measures by means of the used method: Sor (L, R); Mx (L, R); Cd (L, R); Go (L, R); UI; and LI.

The ACM were performed accordingly and provided the betweenness centrality of the values of the go-sagittal plane (Figure 2). The closeness of the links in the following values can be evidenced with N-Me, Mx-Sagittal plane, Cd-Go-Me, S-N-Go-Me, and Cd-Go-N (Figure 3).

## 4. Discussion

A conventional 2D cephalometric analysis using lateral and frontal cephalograms, along with facial scanning, is currently the main diagnostic imaging modality used for orthodontic diagnosis and treatment planning [17,18]. The main disadvantages of the 2D conventional cephalometry are represented by projective displacement, rotational errors, and linear projective transformation, which might affect the reliability and reproducibility of the relevant measurements [19]. In addition, the 2D measurements could be distorted in patients presenting facial asymmetries who need a proper anterio-posterior cephalometry for their treatment plan. However, despite some limitations, conventional cephalometry still represents the golden standard, especially when in the presence of latero-cephalometric radiographs in children and airway monitoring [20]. Some additional information could be obtained, together with asymmetries and jaw width; instead, it is difficult to evaluate the skeletal harmony based on the skull width and length in the lateral view [21].

Frontal cephalometries are subject to inter-operator variations, as many structures are superimposed, making it difficult to accurately identify landmarks and measurements. This problem can be overcome by using CBCT, as the structures can be identified in their 3D organization [7]. Cephalometric measurements using CBCT images have come into use over the past decade and have been found to overcome some of the limitations associated with the traditional cephalometric analysis [22].

One of the advantages of a 3D cephalometric analysis is the ability to provide reliable 3D information while using a single cephalometric analysis and not two or three separate projections. In addition, there is a cautious agreement and consensus among clinicians that the 3D analysis, when compared to conventional cephalometry, could be more accurate and could have a better description of the actual anatomical structures, including a higher reproducibility and better precision [23,24,25,26]. Indeed, both the intra- and inter-rater reliability assessed in the current study were excellent. Many studies have assessed the accuracy and reliability of measurements on CBCT images [2,9,21,27,28]. However, no cephalometric norms for Caucasian patients obtained using CBCT images have been proposed yet in the literature [29].

The development of standard cephalometric norms is important to perform optimal orthodontic and surgical treatment planning above all regarding the mandibular angles, which are difficult to measure in 2D and present more discrepancies [30,31]. In particular, the current study evidenced the necessity of providing 3D norms of the mandibular body curve angles and gonial angles [32]. The values obtained from the 3D analysis were more realistic representations of the actual anatomy without projection problems. An example is represented by the mandibular length (Go left–right Me). The real measurement is calculated, such as in a 3D analysis, on a line that is on an oblique plane relative to the midsagittal plane. In the conventional technique, this measurement is based on a projection leading to a distance that is shorter than the real one. Contrarily, ANB measurements are placed in the midsagittal plane; therefore, the 2D and 3D perspectives result in very similar values [33].

In general, all the measurements that are approximately placed on the midsagittal plane are statistically similar to those obtained in 2D, while the angular measurements not lying on the midsagittal plane show more differences. Additionally, measurements involving mandibular landmarks often show relatively weak correlations. A normality set of data, collected from different groups of Class I patients divided into stages of growth, could give the clinicians a standard reference range to diagnose the malocclusions. Recently, a great number of network software packages have been developed, all of which have their own strengths and weaknesses [34]. In this study, we focused on the use of Pajek, not because of a qualitative choice but because of its immediate availability. Pajek is widely used and relatively inexpensive and is designed to handle very large datasets; in particular, Pajek allows the visualization and simplification of large networks. There are also alternatives to Pajek that might eventually lead to the same results, such as NetDraw and UCINET. Pajek runs on Windows-compatible computers, can be downloaded for free, and is constantly being updated by its developers [35,36]. The software uses spring-embedding algorithms for its layouts based on an assumed attraction between adjacent points (actors that are tied with one another) and an assumed repulsion between nonadjacent points (actors that are not tied to one another) and allocates points in a two-dimensional space.

Ultimately, data mining provided by the use of neural networks helps to understand the network between the values. In our case, we found how the centrality of a set of measurements could be a determinant variable and describe the betweenness centrality.

### Limitations of the Study

There are a few limitations with should be addressed: the study does not take into account other ethnicities rather than Caucasian and should be addressed with caution in different populations. The threshold we set in the middle of Classes II and III should be taken with caution in borderline cases, as a larger sample would help to increase the overall precision. We speculate that an ANN combined with data mining will help clinicians investigate larger samples, especially with the help of fully automated cephalometry.

## 5. Conclusions

This study aims to provide the clinician with a new set of values and their network, which could be used as a reference of normality for 3D cephalometrics distributed among the Caucasian development stages. When compared to 2D values, the 3D technique decreases the risk of underestimating the angular measurements on the lateral landmarks.

Further studies are needed to increase the sample size in order to better define age-related values and their intervals, including scans coming from different centers to increase the data robustness. Moreover, other populations can be evaluated to establish their normal values.

## Figures and Tables

**Figure 1 bioengineering-09-00216-f001:**
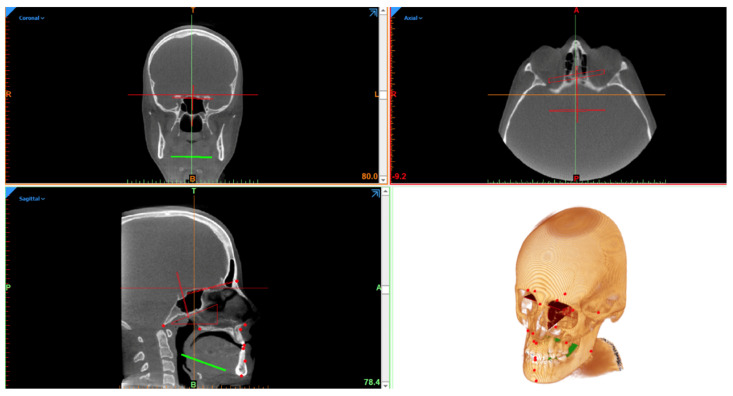
Figure representing the full 3D cephalometry multi-planar vision and three-dimensional reconstruction.

**Figure 2 bioengineering-09-00216-f002:**
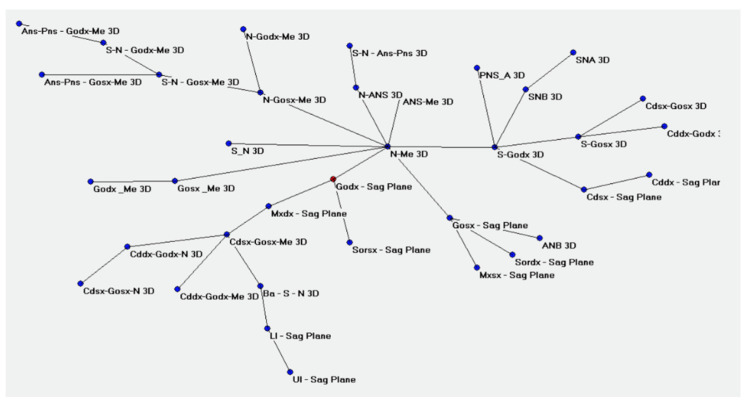
Right go-sagittal plane betweenness centrality displayed on Pajek.

**Figure 3 bioengineering-09-00216-f003:**
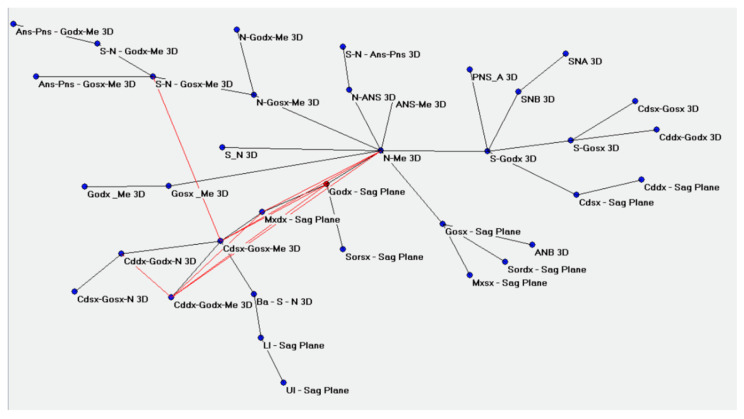
Betweenness centrality displayed on Pajek, calculated with Auto Contractive Maps.

**Table 1 bioengineering-09-00216-t001:** Results of the ICC calculations for the intra-rater reliability. Intra Class Coefficient (ICC), Confidence Interval (CI), Lower (L), and Upper (U) limits; statistical significance (*).

Intra-Rater of Each Observer			Overall			
		N of Measurements	ICC	95% CI		
				LL	UL	*p*
3D	Rater 1	3	1.00	0.997	1.00	<0.001 ***
	Rater 2	3	1.00	0.997	1.00	<0.001 ***
	Rater 3	3	1.00	0.998	1.00	<0.001 ***
2D	Rater 1	3	1.00	0.998	1.00	<0.001 ***
	Rater 2	3	1.00	0.999	1.00	<0.001 ***
	Rater 3	3	1.00	0.999	1.00	<0.001 ***

*** *p* < 0.001.

**Table 2 bioengineering-09-00216-t002:** Descriptive statistics (mean, standard deviation (SD), SEM, and 95% confidence interval (CI) lower and higher) of the 3D and 2D cephalometric measurements and their comparisons (Pearson correlation coefficient R and *p*-value) for the pre-peak age subgroup. Confidence interval (CI), lower (L), upper (U) limits; *p*-values and relevant correlation coefficients are computed by the Pearson correlation analysis; not computed (n/a) statistical significance (*).

	Variables	3D Variables						2D Variables					Comparison	
Orientation	Measurement	Units	Mean	SD	SEM	Lower	Upper	Mean	SD	SEM	Lower	Upper	*p*-Value	R Value
Antero-posterior	S—N	mm	63.66	3.38	0.46	62.73	64.59	63.66	3.41	0.47	62.72	64.60	0.97	1.00
	PNS—A	mm	42.71	3.07	0.42	41.86	43.55	42.69	3.07	0.42	41.85	43.54	0.00 ***	1.00
	GoL—Me	mm	74.83	4.94	0.68	73.47	76.19	62.78	4.70	0.65	61.48	64.08	0.00 ***	0.93
	GoR—Me	mm	75.00	4.71	0.65	73.71	76.30	62.84	4.97	0.68	61.47	64.21	0.00 ***	0.91
Sagittal angular	SNA	deg	80.35	2.86	0.39	79.56	81.14	80.35	2.86	0.39	79.56	81.14	0.79	1.00
	SNB	deg	77.84	2.65	0.36	77.11	78.57	77.83	2.65	0.36	77.10	78.56	0.36	1.00
	ANB	deg	2.60	1.02	0.14	2.32	2.88	2.52	1.04	0.14	2.23	2.80	0.01 **	0.98
Vertical linear	N—Me	mm	101.47	7.46	1.02	99.42	103.53	101.46	7.46	1.02	99.40	103.51	0.02 *	1.00
	N—ANS	mm	45.94	3.64	0.50	44.94	46.95	45.93	3.64	0.50	44.93	46.93	0.01 **	1.00
	ANS—Me	mm	56.72	5.18	0.71	55.29	58.14	56.70	5.18	0.71	55.27	58.13	0.00 ***	1.00
	CdL—GoL	mm	55.77	3.94	0.54	54.68	56.86	49.14	4.24	0.58	47.98	50.31	0.00 ***	0.89
	CdR—GoR	mm	55.87	3.96	0.54	54.78	56.96	49.59	4.36	0.60	48.39	50.79	0.00 ***	0.94
	S—GoL	mm	75.10	5.92	0.81	73.47	76.73	63.26	5.62	0.77	61.71	64.81	0.00 ***	0.97
	S—GoR	mm	75.23	5.50	0.76	73.72	76.75	63.13	5.57	0.77	61.59	64.66	0.00 ***	0.96
Vertical angular	Ba—S—N	deg	130.13	4.94	0.68	128.77	131.49	130.16	4.95	0.68	128.79	131.52	0.00 ***	1.00
	S-N—ANS-PNS	deg	7.94	2.84	0.39	7.16	8.73	7.83	2.87	0.39	7.04	8.62	0.00 ***	1.00
	S-N—GoL-Me	deg	46.90	3.53	0.49	45.92	47.87	35.37	4.21	0.58	34.21	36.53	0.00 ***	0.92
	S-N—GoR-Me	deg	46.84	3.81	0.52	45.79	47.88	35.53	4.27	0.59	34.35	36.70	0.00 ***	0.90
	CdL—GoL—Me	deg	120.85	4.77	0.66	119.53	122.16	123.18	5.99	0.82	121.53	124.83	0.00 ***	0.91
	CdR—GoR—Me	deg	120.66	5.00	0.69	119.28	122.04	123.76	6.62	0.91	121.93	125.58	0.00 ***	0.94
	CdL—GoL—N	deg	46.19	4.58	0.63	44.93	47.46	45.88	4.68	0.64	44.59	47.17	0.00 ***	1.00
	CdR—GoR—N	deg	46.46	4.72	0.65	45.16	47.76	46.22	4.72	0.65	44.92	47.53	0.00 ***	1.00
	N—GoL—Me	deg	65.20	3.74	0.51	64.17	66.23	74.04	5.15	0.71	72.62	75.46	0.00 ***	0.98
	N—GoR—Me	deg	65.07	3.72	0.51	64.04	66.10	74.16	5.15	0.71	72.74	75.58	0.00 ***	0.96
	ANS-PNS—GoL-Me	deg	27.54	4.49	0.62	41.23	43.11	42.17	3.42	0.47	26.30	28.77	0.00 ***	0.84
	ANS-PNS—GoR-Me	deg	41.89	3.61	0.50	40.90	42.89	27.69	4.44	0.61	26.47	28.92	0.00 ***	0.86
Transverse	SorL—Sag Plane	mm	23.31	3.17	0.44	22.44	24.18	23.31	3.17	0.44	22.44	24.18	N/A	N/A
	SorR—Sag Plane	mm	23.35	2.82	0.39	22.58	24.13	23.35	2.82	0.39	22.58	24.13	N/A	N/A
	MxL—Sag Plane	mm	28.24	2.14	0.29	27.65	28.83	28.24	2.14	0.29	27.65	28.83	N/A	N/A
	MxR—Sag Plane	mm	28.25	3.19	0.44	27.37	29.13	28.25	3.19	0.44	27.37	29.13	N/A	N/A
	CdL—Sag Plane	mm	45.15	2.92	0.40	44.34	45.95	45.15	2.92	0.40	44.34	45.95	N/A	N/A
	CdR—Sag Plane	mm	44.81	2.91	0.40	44.01	45.62	44.81	2.91	0.40	44.01	45.62	N/A	N/A
	GoL—Sag Plane	mm	40.69	3.48	0.48	39.73	41.65	40.69	3.48	0.48	39.73	41.65	N/A	N/A
	GoR—Sag Plane	mm	40.44	3.76	0.52	39.40	41.47	40.44	3.76	0.52	39.40	41.47	N/A	N/A
	UI—Sag Plane	mm	2.37	2.06	0.28	1.80	2.94	2.37	2.06	0.28	1.80	2.94	N/A	N/A
	LI—Sag Plane	mm	2.49	2.23	0.31	1.87	3.10	2.49	2.23	0.31	1.87	3.10	N/A	N/A

** *p* < 0.01, *** *p* < 0.001.

**Table 3 bioengineering-09-00216-t003:** Descriptive statistics (mean, standard deviation (SD), SEM, and 95% confidence interval (CI) lower and higher) of the 3D and 2D cephalometric measurements and their comparisons (Pearson correlation coefficient R and *p*-value) for the peak age subgroup. Confidence interval (CI), lower (L), upper (U) limits; *p*-values and relevant correlation coefficients are computed by the Pearson correlation analysis; not computed (n/a) statistical significance (*).

	Variables	3D Variables						2D Variables					Comparison	
Orientation	Measurement	Units	Mean	SD	SEM	Lower	Upper	Mean	SD	SEM	Lower	Upper	*p*-Value	R Value
Antero-posterior	S—N	mm	64.87	3.54	0.58	63.69	66.05	64.89	3.62	0.60	63.69	66.10	0.58	1.00
	PNS—A	mm	44.29	3.18	0.52	43.23	45.35	44.27	3.19	0.52	43.21	45.33	0.00 ***	1.00
	GoL—Me	mm	76.96	5.54	0.91	75.12	78.81	65.35	5.44	0.89	63.54	67.17	0.00 ***	0.96
	GoR—Me	mm	77.58	5.16	0.85	75.86	79.30	64.84	5.54	0.91	62.99	66.69	0.00 ***	0.96
Sagittal angular	SNA	deg	80.84	4.12	0.68	79.47	82.22	78.41	3.82	0.63	77.14	79.69	1.00	1.00
	SNB	deg	80.84	4.13	0.68	79.47	82.22	78.42	3.82	0.63	77.14	79.69	0.58	1.00
	ANB	deg	2.58	1.23	0.20	2.18	2.99	2.49	1.21	0.20	2.09	2.90	0.00 ***	0.99
Vertical linear	N—Me	mm	106.27	7.45	1.23	103.79	108.76	106.25	7.45	1.22	103.76	108.73	0.01 **	1.00
	N—ANS	mm	48.56	3.65	0.60	47.34	49.77	48.54	3.65	0.60	47.32	49.76	0.01 **	1.00
	ANS—Me	mm	58.83	4.95	0.81	57.18	60.48	58.80	4.93	0.81	57.16	60.45	0.00 ***	1.00
	CdL—GoL	mm	52.15	12.84	2.11	47.86	56.43	50.21	5.64	0.93	48.33	52.09	0.23	0.72
	CdR—GoR	mm	51.80	11.92	1.96	47.82	55.77	50.19	5.78	0.95	48.26	52.12	0.22	0.83
	S—GoL	mm	79.84	6.30	1.04	77.74	81.94	67.98	6.47	1.06	65.82	70.13	0.00 ***	0.98
	S—GoR	mm	79.74	6.11	1.01	77.70	81.78	68.15	6.60	1.08	65.95	70.34	0.00 ***	0.97
Vertical angular	Ba—S—N	deg	129.62	6.07	1.00	127.59	131.64	129.65	6.08	1.00	127.62	131.67	0.01 **	1.00
	S-N—ANS-PNS	deg	8.59	3.57	0.59	7.40	9.79	8.45	3.67	0.60	7.23	9.68	0.00 ***	1.00
	S-N—GoL-Me	deg	46.06	3.72	0.61	44.82	47.30	34.36	5.09	0.84	32.67	36.06	0.00 ***	0.92
	S-N—GoR-Me	deg	46.11	4.28	0.70	44.68	47.54	34.43	5.68	0.93	32.54	36.32	0.00 ***	0.93
	CdL—GoL—Me	deg	119.48	9.51	1.56	116.31	122.65	122.45	6.43	1.06	120.30	124.59	0.01 **	0.72
	CdR—GoR—Me	deg	119.47	8.93	1.47	116.49	122.45	122.91	6.53	1.07	120.73	125.09	0.00 ***	0.80
	CdL—GoL—N	deg	54.62	3.95	0.65	53.30	55.94	47.99	3.85	0.63	46.71	49.27	0.00 ***	0.93
	CdR—GoR—N	deg	54.82	3.38	0.56	53.69	55.95	48.09	3.61	0.59	46.88	49.29	0.00 ***	0.89
	N—GoL—Me	deg	66.24	4.70	0.77	64.67	67.80	74.46	4.98	0.82	72.80	76.11	0.00 ***	0.92
	N—GoR—Me	deg	65.75	4.22	0.69	64.34	67.16	74.82	5.47	0.90	72.99	76.64	0.00 ***	0.97
	ANS-PNS—GoL-Me	deg	40.87	2.76	0.45	39.95	41.79	25.93	4.07	0.67	24.57	27.29	0.00 ***	0.79
	ANS-PNS—GoR-Me	deg	40.94	2.91	0.48	39.97	41.91	26.00	4.44	0.73	24.52	27.48	0.01 **	0.71
Transverse	SorL—Sag Plane	mm	24.17	2.74	0.45	23.26	25.08	24.17	2.74	0.45	23.26	25.08	N/A	N/A
	SorR—Sag Plane	mm	24.66	3.35	0.55	23.54	25.77	24.66	3.35	0.55	23.54	25.77	N/A	N/A
	MxL—Sag Plane	mm	29.34	2.53	0.42	28.50	30.19	29.34	2.53	0.42	28.50	30.19	N/A	N/A
	MxR—Sag Plane	mm	29.74	3.55	0.58	28.56	30.93	29.74	3.55	0.58	28.56	30.93	N/A	N/A
	CdL—Sag Plane	mm	46.97	2.48	0.41	46.14	47.80	46.97	2.48	0.41	46.14	47.80	N/A	N/A
	CdR—Sag Plane	mm	46.45	2.72	0.45	45.55	47.36	46.45	2.72	0.45	45.55	47.36	N/A	N/A
	GoL—Sag Plane	mm	41.02	2.97	0.49	40.03	42.01	41.02	2.97	0.49	40.03	42.01	N/A	N/A
	GoR—Sag Plane	mm	41.81	2.78	0.46	40.88	42.74	41.81	2.78	0.46	40.88	42.74	N/A	N/A
	UI—Sag Plane	mm	2.54	1.82	0.30	1.94	3.15	2.54	1.82	0.30	1.94	3.15	N/A	N/A
	LI—Sag Plane	mm	2.47	1.64	0.27	1.93	3.02	2.47	1.64	0.27	1.93	3.02	N/A	N/A

** *p* < 0.01, *** *p* < 0.001.

**Table 4 bioengineering-09-00216-t004:** Descriptive statistics (mean, standard deviation (SD), SEM, and 95% confidence interval (CI) lower and higher) of the 3D and 2D cephalometric measurements and their comparisons (Pearson correlation coefficient R and *p*-value) for the post-peak age subgroup. Confidence interval (CI), lower (L), upper (U) limits; *p*-values and relevant correlation coefficients are computed by the Pearson correlation analysis; not computed (n/a) statistical significance (*).

	Variables	3D Variables						2D Variables					Comparison	
Orientation	Measurement	Units	Mean	SD	SEM	Lower	Upper	Mean	SD	SEM	Lower	Upper	*p*-Value	R Value
Antero-posterior	S—N	mm	66.92	4.74	0.95	64.97	68.88	67.00	4.86	0.97	64.99	69.01	0.17	1.00
	PNS—A	mm	45.98	3.28	0.66	44.62	47.33	45.97	3.28	0.66	44.62	47.33	0.00 ***	1.00
	GoL—Me	mm	81.81	4.24	0.85	80.06	83.56	69.32	4.50	0.90	67.46	71.18	0.00 ***	0.91
	GoR—Me	mm	82.48	4.86	0.97	80.47	84.49	69.25	5.44	1.09	67.00	71.49	0.00 ***	0.94
Sagittal angular	SNA	deg	80.20	2.77	0.55	79.06	81.34	80.20	2.77	0.55	79.06	81.35	0.55	1.00
	SNB	deg	78.40	2.90	0.58	77.20	79.60	78.40	2.90	0.58	77.20	79.60	0.41	1.00
	ANB	deg	2.30	0.88	0.18	1.94	2.66	2.03	0.93	0.19	1.65	2.41	0.02 *	0.83
Vertical linear	N—Me	mm	114.27	7.40	1.48	111.22	117.33	114.25	7.39	1.48	111.20	117.30	0.00 ***	1.00
	N—ANS	mm	51.59	2.99	0.60	50.36	52.83	51.58	2.99	0.60	50.34	52.81	0.00 ***	1.00
	ANS—Me	mm	63.61	5.45	1.09	61.36	65.85	63.57	5.44	1.09	61.33	65.81	0.01 **	1.00
	CdL—GoL	mm	55.05	7.33	1.47	52.03	58.08	54.86	7.34	1.47	51.83	57.89	0.83	0.94
	CdR—GoR	mm	55.16	6.99	1.40	52.28	58.05	54.99	6.98	1.40	52.12	57.87	0.00 ***	0.87
	S—GoL	mm	85.16	9.04	1.81	81.43	88.89	72.84	9.54	1.91	68.90	76.78	0.00 ***	0.98
	S—GoR	mm	85.44	8.25	1.65	82.03	88.84	73.13	8.91	1.78	69.45	76.80	0.00 ***	0.99
Vertical angular	Ba—S—N	deg	129.70	6.18	1.24	127.15	132.25	129.71	6.18	1.24	127.15	132.26	0.21	1.00
	S-N—ANS-PNS	deg	8.61	3.02	0.60	7.36	9.85	8.53	3.08	0.62	7.26	9.81	0.02 *	1.00
	S-N—GoL-Me	deg	46.78	4.25	0.85	45.02	48.53	35.64	6.09	1.22	33.13	38.16	0.00 ***	0.93
	S-N—GoR-Me	deg	46.91	4.53	0.91	45.04	48.78	35.55	5.93	1.19	33.11	38.00	0.00 ***	0.93
	CdL—GoL—Me	deg	118.84	4.74	0.95	116.88	120.79	122.29	6.08	1.22	119.78	124.80	0.00 ***	0.91
	CdR—GoR—Me	deg	118.22	5.21	1.04	116.07	120.37	122.18	7.09	1.42	119.25	125.10	0.00 ***	0.93
	CdL—GoL—N	deg	51.72	4.45	0.89	49.88	53.55	45.98	4.77	0.95	44.02	47.95	0.00 ***	0.93
	CdR—GoR—N	deg	51.56	4.04	0.81	49.89	53.23	45.86	4.60	0.92	43.96	47.76	0.00 ***	0.93
	N—GoL—Me	deg	67.35	3.75	0.75	65.80	68.89	76.30	4.85	0.97	74.30	78.31	0.00 ***	0.96
	N—GoR—Me	deg	67.06	4.18	0.84	65.34	68.79	76.32	5.66	1.13	73.98	78.66	0.00 ***	0.97
	ANS-PNS—GoL-Me	deg	41.60	3.44	0.69	40.18	43.02	27.18	5.81	1.16	24.79	29.58	0.00 ***	0.83
	ANS-PNS—GoR-Me	deg	41.54	4.40	0.88	39.73	43.36	27.09	5.64	1.13	24.76	29.42	0.00 ***	0.87
Transverse	SorL—Sag Plane	mm	24.93	3.36	0.67	23.54	26.32	24.93	3.36	0.67	23.54	26.32	N/A	N/A
	SorR—Sag Plane	mm	25.71	4.07	0.81	24.03	27.38	25.71	4.07	0.81	24.03	27.38	N/A	N/A
	MxL—Sag Plane	mm	29.32	2.50	0.50	28.29	30.35	29.32	2.50	0.50	28.29	30.35	N/A	N/A
	MxR—Sag Plane	mm	28.94	3.20	0.64	27.62	30.26	28.94	3.20	0.64	27.62	30.26	N/A	N/A
	CdL—Sag Plane	mm	47.28	2.98	0.60	46.05	48.51	47.28	2.98	0.60	46.05	48.51	N/A	N/A
	CdR—Sag Plane	mm	47.09	2.59	0.52	46.02	48.16	47.09	2.59	0.52	46.02	48.16	N/A	N/A
	GoL—Sag Plane	mm	44.10	3.53	0.71	42.64	45.55	44.10	3.53	0.71	42.64	45.55	N/A	N/A
	GoR—Sag Plane	mm	43.77	3.21	0.64	42.44	45.09	43.77	3.21	0.64	42.44	45.09	N/A	N/A
	UI—Sag Plane	mm	1.69	1.66	0.33	1.01	2.38	1.69	1.66	0.33	1.01	2.38	N/A	N/A
	LI—Sag Plane	mm	1.94	1.86	0.37	1.17	2.71	1.94	1.86	0.37	1.17	2.71	N/A	N/A

** *p* < 0.01, *** *p* < 0.001.

**Table 5 bioengineering-09-00216-t005:** Normality range divided into the age subgroups.

	Variables		Pre-Peak			Peak			Post-Peak		
Orientation	Measurement	Units	Normality			Normality			Normality		
Antero-posterior	S—N	mm	63.66	±	3.38	64.87	±	3.54	66.92	±	4.74
	PNS—A	mm	42.71	±	3.07	44.29	±	3.18	45.98	±	3.28
	GoL—Me	mm	74.83	±	4.94	76.96	±	5.54	81.81	±	4.24
	GoR—Me	mm	75.00	±	4.71	77.58	±	5.16	82.48	±	4.86
Sagittal angular	SNA	deg	80.35	±	2.86	80.84	±	4.12	80.20	±	2.77
	SNB	deg	77.84	±	2.65	80.84	±	4.13	78.40	±	2.90
	ANB	deg	2.60	±	1.02	2.58	±	1.23	2.30	±	0.88
Vertical linear	N—Me	mm	101.47	±	7.46	106.27	±	7.45	114.27	±	7.40
	N—ANS	mm	45.94	±	3.64	48.56	±	3.65	51.59	±	2.99
	ANS—Me	mm	56.72	±	5.18	58.83	±	4.95	63.61	±	5.45
	CdL—GoL	mm	55.77	±	3.94	52.15	±	12.84	55.05	±	7.33
	CdR—GoR	mm	55.87	±	3.96	51.80	±	11.92	55.16	±	6.99
	S—GoL	mm	75.10	±	5.92	79.84	±	6.30	85.16	±	9.04
	S—GoR	mm	75.23	±	5.50	79.74	±	6.11	85.44	±	8.25
Vertical angular	Ba—S—N	deg	130.13	±	4.94	129.62	±	6.07	129.70	±	6.18
	S-N—ANS-PNS	deg	7.94	±	2.84	8.59	±	3.57	8.61	±	3.02
	S-N—GoL-Me	deg	46.90	±	3.53	46.06	±	3.72	46.78	±	4.25
	S-N—GoR-Me	deg	46.84	±	3.81	46.11	±	4.28	46.91	±	4.53
	CdL—GoL—Me	deg	120.85	±	4.77	119.48	±	9.51	118.84	±	4.74
	CdR—GoR—Me	deg	120.66	±	5.00	119.47	±	8.93	118.22	±	5.21
	CdL—GoL—N	deg	46.19	±	4.58	54.62	±	3.95	51.72	±	4.45
	CdR—GoR—N	deg	46.46	±	4.72	54.82	±	3.38	51.56	±	4.04
	N—GoL—Me	deg	65.20	±	3.74	66.24	±	4.70	67.35	±	3.75
	N—GoR—Me	deg	65.07	±	3.72	65.75	±	4.22	67.06	±	4.18
	ANS-PNS—GoL-Me	deg	27.54	±	4.49	40.87	±	2.76	41.60	±	3.44
	ANS-PNS—GoR-Me	deg	41.89	±	3.61	40.94	±	2.91	41.54	±	4.40
Transverse	SorL—Sag Plane	mm	23.31	±	3.17	24.17	±	2.74	24.93	±	3.36
	SorR—Sag Plane	mm	23.35	±	2.82	24.66	±	3.35	25.71	±	4.07
	MxL—Sag Plane	mm	28.24	±	2.14	29.34	±	2.53	29.32	±	2.50
	MxR—Sag Plane	mm	28.25	±	3.19	29.74	±	3.55	28.94	±	3.20
	CdL—Sag Plane	mm	45.15	±	2.92	46.97	±	2.48	47.28	±	2.98
	CdR—Sag Plane	mm	44.81	±	2.91	46.45	±	2.72	47.09	±	2.59
	GoL—Sag Plane	mm	40.69	±	3.48	41.02	±	2.97	44.10	±	3.53
	GoR—Sag Plane	mm	40.44	±	3.76	41.81	±	2.78	43.77	±	3.21
	UI—Sag Plane	mm	2.37	±	2.06	2.54	±	1.82	1.69	±	1.66
	LI—Sag Plane	mm	2.49	±	2.23	2.47	±	1.64	1.94	±	1.86

## Data Availability

Not applicable.

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
