# Peer review of "3D Cephalometric Normality Range: Auto Contractive Maps (ACM) Analysis in Selected Caucasian Skeletal Class I Age Groups"

_bioengineering, 2022, doi:10.3390/bioengineering9050216_

Round 1
Reviewer 1 Report
Dear Authors, I read your article with interest. However, there are some aspects to be corrected:
- Please correct authors affiliations. It must be formatted correctly following the journal author guidelines.
- The abstract should not exceed 200 words, in your case 272 words were used. Please shorten the abstract
- Too many self-citations should not be used. The name M. Ferronato appears 4 times in the bibliography.
- The introduction should be enriched with content. I recommend adding some references concerning ANN and ACM.
- I found the materials and methods section a bit confusing. Please better divide the various paragraphs (2.1, 2.2, etc).
- Page 4 regarding materials and methods is too confused and needs to be modified by the authors to better understand the text.
- In all tables, authors are encouraged to emphasize the statistical significance with an asterisk (e.g.- p<0.01*).
Author Response
Dear Authors, I read your article with interest. However, there are some aspects to be corrected:
Dear Reviewer, thanks for taking the time to read our manuscript, we are grateful for your kind comments and opinions, below you will find our answers to each point.
- Please correct authors affiliations. It must be formatted correctly following the journal author guidelines.
Dear reviewer, we have reformatted the authors’ affiliations. Kindly let us know if it is correct now.
- The abstract should not exceed 200 words, in your case 272 words were used. Please shorten the abstract
Thanks, we have reduced the total number of words in the abstract. It is under 200 words now.
- Too many self-citations should not be used. The name M. Ferronato appears 4 times in the bibliography.
Thanks, we removed citation 32; 34, we believe the others are strictly related to 3D cephalometry (22: 3D cephalometry on reduced FOV CBCT: skeletal class assessment through AF-BF on Frankfurt plane-validity and reliability through comparison with 2D measurements; 17: A comparison between stereophotogrammetry and smartphone structured light technology for three-dimensional face scanning.) and have a key-role in our manuscript; we kindly ask you to understand that self-citation practice is common when researchers/institutions are investigating on a certain topic, specific to their research path, please consider this was not intended as a misuse.
- The introduction should be enriched with content. I recommend adding some references concerning ANN and ACM.
Thanks for the comments, we added a section in the introduction.
- I found the materials and methods section a bit confusing. Please better divide the various paragraphs (2.1, 2.2, etc).
Thanks for your kind suggestion, the Materials and Methods section is now divided into the paragraphs: 2.1 Patient selection; 2.2 Age and sex distribution; 2.3 Scanning protocol; 2.4 Data elaboration; 2.5 3D Cephalometrics; 2.6 Data reliability; 2.7 Statistical analysis; kindly let us know if it looks better now.
- Page 4 regarding materials and methods is too confused and needs to be modified by the authors to better understand the text.
Dear reviewer, we re-wrote the corresponding section accordingly, please find it now under the subparagraph 2.5 and kindly tell us if it is more clear now.
- In all tables, authors are encouraged to emphasize the statistical significance with an asterisk (e.g.- p<0.01*).
Thanks, we revised accordingly.
Reviewer 2 Report
" 3D cephalometric normality range: Auto Contractive Maps (ACM) analysis in selected caucasian skeletal Class I age groups"
It is very interesting to define normal values for caucasian age groups of a novel 3D cephalometric analysis and to define the links through Artificial Neural Network using Pajek. However, there are a few corrections that are essential to meet the standard for publication. Please refer to the following comments.
1) There are few explanations and discussions about "Pajek", which is the core of this research. Please add the version, operating environment, and evaluation method.
Also, please add to the discussion section about the advantages of using Pajek in this study.
2) Please add the prospects and limitations of this study to the discussion section. In particular, please describe how to use Pajek in clinical practice and future developments regarding the outlook.
3) Many of the references in this manuscript are old. Please update to the latest literature as much as possible.
Author Response
" 3D cephalometric normality range: Auto Contractive Maps (ACM) analysis in selected caucasian skeletal Class I age groups"
It is very interesting to define normal values for caucasian age groups of a novel 3D cephalometric analysis and to define the links through Artificial Neural Network using Pajek. However, there are a few corrections that are essential to meet the standard for publication. Please refer to the following comments.
Dear Reviewer, thanks for your kind interest manifested towards our manuscript, we are pleased to give you a list of answers and modifications given based on your comments.
1) There are few explanations and discussions about "Pajek", which is the core of this research. Please add the version, operating environment, and evaluation method.
Also, please add to the discussion section about the advantages of using Pajek in this study.
Dear reviewer, thanks for your kind suggestions, we added some key citations and a whole new section in the discussion part to better highlight the utility, version and advantages of using Pajek, a freeware software which is widely used in the network analysis area. Please find our addictions through the text. Pajek can be considered a reference standard when referring to the visualization of networks, we added some parameters, but we think that adding too many technical explanation would be out of the scope of the research and finally disorient the dental/orthodontic clinician readers, the main target of this special issue.
2) Please add the prospects and limitations of this study to the discussion section. In particular, please describe how to use Pajek in clinical practice and future developments regarding the outlook.
Dear reviewer thanks for your kind comment, we added the correct instructions to use Pajek and to correctly read a *.net file, we added a limitations paragraph in the discussion along with speculations on future implementations kindly let us know if this correctly reflects your suggestions.
3) Many of the references in this manuscript are old. Please update to the latest literature as much as possible.
Dear reviewer, thanks for your kind punctuation, we removed the oldest references and revised substantially all the citations used.
Reviewer 3 Report
thanks for the article, and I have the following comments:
- In abstract and title, you have mentioned ANN and ACM, so in the M&M, how to establish the algorithm of ANN or ACM? I can only see the landmark identification by operators, so how to validate ANN or ACM? how many test sets and how many training sets?
- Pajek is a large network analysis software so how does it relate to ACM or ANN?
- In your discussion, you have mentioned conventional 2D Cephalometric analysis. Despite it has certain limitations, it is still one of the gold standards for orthodontics. If you transform CBCT to 2D, the image distortion would also be great. Have you consider that? I believe you should downtone the first paragraph of Discussion and also mention some useful advantages for 2D cephalometric analysis, e.g. useful in LCR in children ( Angle Orthod (2020) 90 (1): 47–55. ) and airway monitoring (10.1111/ocr.12442). In terms of skull width, ref [21] is a CBCT study that did not clinically reflect the case - the CCD in this article (https://doi.org/10.1259/dmfr.20210015 ) maybe is a better and newer article for you to cite.
- For CBCT, besides the landmark, you have to write some more information in discission why it is valuable, e.g. monitoring of mid palatal sutures (Scientific Reports 12 (1), 1-8 ) and identifications of foramini ( ). These extra information would be of valuable of treatment planning , unlike the 2D cephalometric images.
Author Response
thanks for the article, and I have the following comments:
Dear reviewer, we thank you for agreeing to revise and giving your comments to our manuscript, you will find our comments listed below:
In abstract and title, you have mentioned ANN and ACM, so in the M&M, how to establish the algorithm of ANN or ACM?
Dear reviewer, please cross-check the answer given to the peer referees #1 and #2, please find the complete description in the bibliography under the [12] reference, it is out of our scopes and competences to give a full description on how the algorithms were composed, we are out of the core team behind the development of those systems and we based our research on the UX given and by the instructions, guidelines and results available and retrieved from the literature. Please, kindly let us know if you want us to extract the algorithms behind, but we are not sure this is of interest to the orthodontists readers, the target of this study and of the Special Issue. We hope our instructions and references are viable to scientifically reproduce the results achieved.
I can only see the landmark identification by operators, so how to validate ANN or ACM? how many test sets and how many training sets?
Dear reviewer, ACM is a previously validated automatic method, we added the settings we used and further descriptions on the softwares used, kindly let us know if you find those informations useful, thanks.
Pajek is a large network analysis software so how does it relate to ACM or ANN?
Dear reviewer we added a full description on software usage through the test in the Introduction, Materials and methods and discussion, kindly let us know if this clears your doubts.
In your discussion, you have mentioned conventional 2D Cephalometric analysis. Despite it has certain limitations, it is still one of the gold standards for orthodontics. If you transform CBCT to 2D, the image distortion would also be great. Have you consider that? I believe you should downtone the first paragraph of Discussion and also mention some useful advantages for 2D cephalometric analysis, e.g. useful in LCR in children ( Angle Orthod (2020) 90 (1): 47–55. ) and airway monitoring (10.1111/ocr.12442).
Dear reviewer, thanks for pointing this out, we are biased and very grateful for evidencing this, we modified the respective paragraph and down toned the discussion, also we added your reference suggestion, kindly revise please.
In terms of skull width, ref [21] is a CBCT study that did not clinically reflect the case - the CCD in this article (https://doi.org/10.1259/dmfr.20210015 ) maybe is a better and newer article for you to cite.
Dear reviewer, thanks again for your kind suggestion, this is very useful for us as helps to update the bibliographic references, a point also evidenced by the other reviewers.
For CBCT, besides the landmark, you have to write some more information in discission why it is valuable, e.g. monitoring of mid palatal sutures (Scientific Reports 12 (1), 1-8 ) and identifications of foramini (10.11607/prd.4770). These extra information would be of valuable of treatment planning , unlike the 2D cephalometric images.
Dear reviewer, thanks for your kind suggestion, please kindly acknowledge that this manuscript has the main goal of evidencing 3D cephalometry, the differences between CBCT and 2D radiographs are a very wide topic, we cannot add to many citations for a single author (Savoldi et al.) since this could be a source of bias. This is fully described in the Cochrane collaboration tool (https://doi.org/10.1136/bmj.d5928) and is generally classified under Reporting bias as a selective reporting. Thanks for your kind comprehension.
Round 2
Reviewer 2 Report
Thank you for giving me this opportunity to re-review your revised manuscript.
I am happy that all of the suggested corrections have been made.
Thank you for spending so much time for revised manuscript.